# A New Method of Inland Water Ship Trajectory Prediction Based on Long Short-Term Memory Network Optimized by Genetic Algorithm

**Long Qian** [1,2]**, Yuanzhou Zheng** [1,2,]*****, Lei Li** [1,2]**, Yong Ma** [1,2]**, Chunhui Zhou** [1,2] **and Dongfang Zhang** [3]

[1] School of Navigation, Wuhan University of Technology, Wuhan 430036, China; lqian@whut.edu.cn (L.Q.); 246968@whut.edu.cn (L.L.); myongdl@whut.edu.cn (Y.M.); chunhui@whut.edu.cn (C.Z.)

[2] Hubei Key Laboratory of Inland Shipping Technology, Wuhan University of Technology, Wuhan 430036, China

[3] Suzhou Port and Shipping Development Center, Suzhou 215000, China; dancywang0414@163.com

***** Correspondence: yzzheng@whut.edu.cn; Tel.: +86-18672966230

**Abstract:** Ship position prediction plays a key role in the early warning and safety of inland waters and maritime navigation. Ship pilots must have in-depth knowledge of the future position of their ship and target ship in a specific time period when maneuvering the ship to effectively avoid collisions. However, prediction accuracy and computing efficiency are crucial issues that need to be worked out at present. To solve these problems, in this paper, the deep long short-term memory network framework (LSTM) and genetic algorithm (GA) are introduced to predict the ship trajectory of inland water. Firstly, the collected actual automatic identification system (AIS) data are preprocessed and a series of typical trajectories are extracted from them; then, the LSTM network is used to predict the typical trajectories in real time. Considering that the hyperparameters of the LSTM network have difficulty obtaining the optimal solution manually, the GA is used to optimize hyperparameters of LSTM; finally, the GA-LSTM trajectory prediction model is constructed with the optimal network parameters and compared with the traditional support vector machine (SVM) model and LSTM model. The experimental results show that the GA-LSTM model effectively improves the accuracy and speed of trajectory prediction, with outstanding performance and good generalization, which possess certain reference values for the development of collision avoidance of unmanned ships.

**Keywords:** trajectory prediction; inland water; LSTM; GA; SVM

## 1. Introduction

"A country with strong transportation, parallel by land and water." As one of the comprehensive transportation modes, waterway transportation undertakes nearly 90% of the world's bulk trade freight volume. Due to its characteristics of low cost and large volume, the activity of the waterway transportation industry is regarded as a barometer of world and regional economic recovery. [1]. With the rapid development of China's shipping industry, the number of ships in the oceans, especially inland rivers, has increased dramatically, leading to an increasing trend of ship traffic accidents. Therefore, in ports or waters with high traffic density and complex conditions, improving the safety of ship navigation is a key issue [2,3]. Vessel Traffic Service (VTS) [4,5] can accurately and effectively monitor and predict the real-time trajectory of ships, which provides a technical support for the early warning of marine traffic accidents. In order to improve the safety of ship navigation in inland river environments, it is necessary to provide real-time trajectory prediction and risk warning functions for the ship's intelligent navigation system. However, the inland river environment is complex and changeable, and traffic accidents are prone to occur, especially in crowded docks and bridge areas. Consequently, it is difficult to predict the trajectory of ships [6].

In recent years, domestic and foreign scholars have proposed a series of ship trajectory prediction models. Anderson [7] takes time as the independent variable, obtains the measured value of the trajectory in discrete time, and regards the trajectory as a one-dimensional Gaussian process. A prior continuous time is defined by a nonlinear time-varying stochastic differential equation driven by white noise. By obtaining the joint prior density and covariance matrix of the observed and the predicted value, the posterior distribution of the predicted value is calculated, and the smoothing trajectory is predicted by combining with dynamics. This method is computationally intensive, and its accuracy gradually decreases over time. Jiang [8] proposed the polynomial Kalman filter method to fit the ship trajectory. This method implements trajectory prediction in a recursive manner, which occupies less memory space in the calculation process and can achieve short-term prediction. However, the assumptions of initial state and ideal conditions of the model have a greater impact on the prediction results. Literature [9] divides the specified sea area into grids, calculates the grid state with the ship's position, speed, and direction as the key factors, and uses the K-order Markov chain to establish a state transition matrix for prediction; however, the utilization rate of historical track information is poor when calculating the grid state at each moment. Zhang [10] proposed a spatial clustering method based on hierarchical density clustering, adopting the DBSCAN model to cluster and denoise the original AIS trajectories to achieve the purpose of predicting ship trajectories. Rong [11] proposed a new probabilistic trajectory prediction model, which described the uncertainty of the future position of the ship trajectory through a continuous probability distribution and has high prediction accuracy.

With the continuous in-depth research of artificial neural networks (ANN) [12,13], the ship trajectory prediction model based on ANN is becoming more and more popular and is widely used in the field of ship navigation [14–16]. Literature [17] uses the back-propagation (BP) neural network model to train and predict ship trajectory with longitude, latitude, and speed information in the AIS data, but the BP neural network has a weak ability to deal with nonlinear problems, and only in the case of a short track are the prediction results more accurate. Literature [18] uses a support vector machine (SVM) to establish a ship trajectory prediction model and adopts speed over ground (SOG), course over ground (COG), longitude (LON), latitude (LAT), and time stamp as the input sample features, which improves the prediction efficiency and accuracy to a certain extent; however, SVM has shortcomings such as weak generalization ability and ease of falling into local extreme values. Brian [19] proposed a dual linear auto-encoder method to predict the future trajectories of selected ships. The auto-encoder consists of two modules, encoding and decoding, which can extract hidden features of AIS data, and the model can predict the trajectory of multiple ships. However, in the process of trajectory features extraction, useless data features cannot be effectively filtered, so the prediction effect of the model is poor. Mao [20] proposed a method for predicting ship trajectory based on an extreme learning machine (ELM). As a single-hidden-layer feed-forward neural network model, the ELM does not require weights and biases of the iterative network and has a high calculation speed. However, the number of hidden-layer nodes in the ELM model is difficult to determine, which affects the generalization performance of the network.

Recurrent neural networks (RNNs) have been extensively developed due to their powerful ability to process sequence information and predictable time information. Hochreater et al. [21] improved the RNN unit structure and proposed a long short-term memory network (LSTM) model, which solved the problems of gradient disappearance, gradient explosion, and insufficient information memory ability by designing the "gate" structure, and LSTM networks can effectively use long-distance timing information [22]. LSTM networks have been successfully applied in speech recognition [23], text processing [24], and other fields, yet there are some defects key hyperparameters, such as the number of hidden-layer neurons, learning rate, etc., which are difficult to determine [25]. Because the number of hidden-layer neurons plays a decisive role in the fitting ability of the model, the learning rate directly affects the convergence speed and calculation time of the model and

the topology of the model is controlled directly by the LSTM network structure parameters. Therefore, the prediction performance of the model established by different hyperparameters is quite different, and how to select the appropriate parameters is very important for the establishment of the model. At present, the hyperparameters of the network model are often selected based on the experience of the researchers and the results of multiple experiments. The randomness is relatively large, which affects the prediction performance of the model to a certain extent.

In order to predict the ship trajectory quickly and accurately, this paper adopts the LSTM network model [26–28] as the technical basis to establish an inland river ship trajectory prediction model. Considering that the key hyperparameters of the current LSTM model are difficult to determine, such as the number of hidden-layer neurons, learning rate, etc., the genetic algorithm (GA) is proposed to optimize the key hyperparameters of LSTM networks. The model takes LOG, LAT, SOG, and COG as the input features and the future position of the ship as the target output. The LSTM neural network model optimized by the GA (GA-LSTM) is used to predict the ship trajectory. The experimental results show that, compared with the current classical LSTM and SVM, the algorithm proposed in this paper can predict the ship trajectory more quickly and accurately to a certain extent.

The remainder of this paper is organized as follows. Section 2 describes the ship trajectory prediction model. Section 3 describes the theoretical background of the LSTM, GA and GA-LSTM models. Moreover, Section 4 mainly contains experiments and analysis. Finally, Section 5 concludes the paper.

## 2. Ship Trajectory Prediction Model

An automatic identification system (AIS) can provide real-time ship trajectory data for detecting the navigation status of ships. Nowadays, it is widely used in ship collision avoidance, maritime monitoring, ship traffic flow forecasting, and maritime accident investigation mechanisms [29]. When a ship is sailing, it mainly relies on the AIS data from the target ship to obtain its navigation behavior, so as to make timely and accurate collision avoidance decisions in complex encounters. In actual navigation, the navigation behavior of a ship is mainly reflected in the changes of characteristic variables such as ship position, SOG, and COG [30]. It is assumed that the navigation behavior of a ship at time $t$ can be characterized as:

$$y(t) = \{lat_t, lon_t, sog_t, cog_t\} \tag{1}$$

where $lon_t$, $lat_t$, $sog_t$, and $cog_t$ are respectively LON, LAT, SOG, and COG of the ship at time $t$.

Generally speaking, the navigation behavior of the ship at the next moment is the result of the current behavior and historical behavior. Therefore, in order to improve the accuracy of the model, the navigation behavior of the ship at the past three moments, $y(t-2)$, $y(t-1)$, and $y(t)$, is taken as the input of the model and the LON and LAT of the ship at the next moment as the output of the model, namely:

$$\begin{cases} I_{input} = \{y(t-2), y(t-1), y(t)\} \\ O_{output} = \{lat_{t+1}, lon_{t+1}\} \end{cases} \tag{2}$$

This is the functional relationship between $I_{input}$ and $O_{output}$: $O_{output} = f(I_{input})$ where $f(\cdot)$ is the nonlinear transformation function.

Therefore, for ship trajectory prediction, with some samples of AIS data as the training data set $\{[y(t-2), y(t-1), y(t)], y(t+1), t = 1, 2, \dots, l\}$, obtaining the best estimate of the nonlinear transformation between the input sample $I_{input}$ and the target output sample $O_{output}$ is a problem. This paper selects the GA-LSTM model to fit the nonlinear transformation function $f(\cdot)$, the GA-LSTM model is constructed with training data, and then the test data are input into the GA-LSTM model, and finally the real-time prediction of the ship trajectory is carried out.

## 3. LSTM Network Optimized by GA

### 3.1. LSTM Network Model

The LSTM neural network solves the problem of gradient disappearance and the explosion of traditional recursive neural networks linked by network units in a chain way, which can effectively improve the learning time. In dealing with the prediction of time series and nonlinear mapping problems, the LSTM model with memory ability shows strong advantages [22]. A structure called a memory cell is added to LSTM to memorize past information, and three gate structures, input gate, output gate, and forget gate, are added to control the transmission of historical information [31].

The structure of the LSTM neural network is shown in Figure 1. Supposing that the network input is $(x_1, x_2, \ldots, x_T)$ and the hidden-layer state is $(h_1, h_2, \ldots, h_T)$, at time t, the calculations of each unit and gate are shown in Equations (3)–(8):

$$i_t = \sigma(w_i \cdot [h_{t-1}, x_t] + b_i) \tag{3}$$

$$f_t = \sigma(w_f \cdot [h_{t-1}, x_t] + b_f) \tag{4}$$

$$\widetilde{c}_t = \tanh(w_c \cdot [h_{t-1}, x_t] + b_c) \tag{5}$$

$$c_t = f_t \circ c_{t-1} + i_t \circ \widetilde{c}_t \tag{6}$$

$$o_t = \sigma(w_o \cdot [h_{t-1}, x_t] + b_o) \tag{7}$$

$$h_t = o_t \circ \tanh(c_t) \tag{8}$$

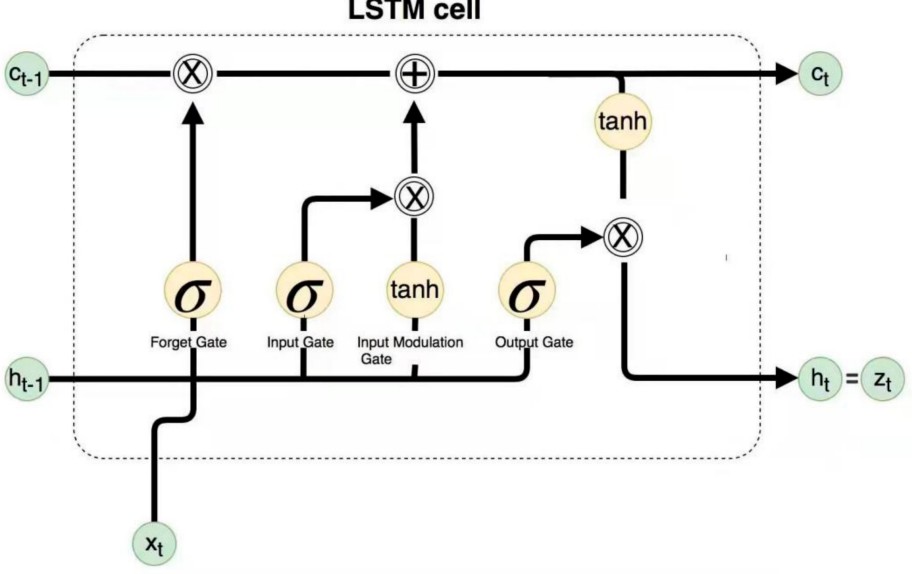

**Figure 1.** The LSTM neural network structure.

In Formulas (3)–(8), $i_t$, $f_t$, and $o_t$ are the calculations of input gate, forget gate, and output gate, respectively; among them, the input gate is mainly used to determine how much input information at the current moment is retained to the unit state at the current moment; the forget gate is mainly used to determine how much information of the unit state $c_{t-1}$ from the previous moment is retained in the current cell state $c_t$; the output gate is mainly used to determine how much output the current cell state has. $h_t$ is the final output of the network, $\widetilde{c}_t$ is the current input unit state; $c_t$ is the current moment unit state; $w_i$, $w_f$, $w_c$, and $w_o$ are the weight matrices of the three gates and unit states; $b_i$, $b_f$, $b_c$, and $b_o$ are respectively the bias of each gate and unit state; $\sigma(\cdot)$ and $\tanh(\cdot)$ are transfer functions; $\cdot$ represents the vector inner product; and the symbol $\circ$ represents element-wise multiplication.

### 3.2. Genetic Algorithm

The GA is usually a biological scientific algorithm that simulates Darwin's theory of biological evolution by a computer, proposed by J. Holland [32] in 1975. In the genetic evolution of populations in the GA, it is found that chromosomes are used as the main carrier of population inheritance, and with the help of a variety of random operations—gene selection, gene crossover, and gene mutation—a new solution set population is constantly evolving. According to the value of individual fitness and the selection function, the optimal population individual can be selected, which is the optimal solution of the optimization problem in the GA.

In this paper, the GA is used to optimize the key hyperparameters of the LSTM network and the powerful global random search ability of the GA is adopted to obtain the optimal combination of the number of neurons and the learning rate in the LSTM network. The basic idea is as follows:

(1)　Chromosome coding

The number of hidden-layer neurons and the learning rate in the LSTM network are taken as the initialization objects of the GA, and chromosome coding is carried out in the form of real-number coding. The interval range of hidden-layer neurons is set to [2, 40], and the interval range of the learning rate is set to [0.001–0.1].

(2)　Fitness function

The fitness function is applied to determine which individuals in the population can perform next-generation genetic operations. According to differing individual fitness, the "survival of the fittest" is used for screening individuals. The selection-of-fitness function directly affects the performance of the optimized network by the GA and then affects the performance of prediction. This paper mainly constructs the fitness function based on the overall fit between the estimated value and the true value of the ship navigation position. In order to make the network parameters obtained by the GA more suitable for the LSTM model and improve the generalization ability of the model, the AIS data are divided into training samples and test samples. The training samples are utilized for LSTM network training. After reaching the limit of the number of iterations, the training sample output value and test sample output value of the LSTM network are obtained. Then the individual fitness function is defined as:

$$fitness = 0.5 \times \frac{1}{J} \sum_{j=1}^{J} \left( \widehat{y}^j{}_t - y_t^j \right)^2 + 0.5 \times \frac{1}{K} \sum_{k=1}^{K} \left( \widehat{y}^k{}_v - y_v^k \right)^2 \tag{9}$$

where $\widehat{y}^j{}_t$ and $\widehat{y}^k{}_v$ are the predicted value of the training sample and the predicted value of the test sample, respectively, and $y_t^j$ and $y_v^k$ are the actual value. The error of the test sample directly reflects the prediction effect of the model; therefore, the fitness function *fitness* includes not only the fitting error of the training sample but also the verification error of the test sample. In the experiment, the error of the training sample and the error of the test sample are given the same weight, which is 0.5, and the sum of the two multiplied with the weight is used as the fitness function of the model.

(3)　Selection operator, crossover operator, and mutation operator

The selection operator selects individuals with better adaptability as parents in the current population and passes genetic information to the offspring. Here, the tournament selection algorithm is used as the GA selection strategy. This selection strategy has the characteristics of efficient algorithm execution rate and easy implementation, and its algorithm complexity is much lower than other selection strategies and is easy to parallelize. It is not easy to fall into the local optimal individual during the selection process and does not require sorting the fitness values of all individuals. The crossover operator takes the shuffle crossover algorithm. Before the crossover, the random.shuffle function is used to perform the shuffle operation in the parent, and then when the random number generated between

0 and 1 is less than the given crossover rate, the crossover transformation is performed. In the mutation operator, when the random number generated between 0 and 1 is less than the given mutation rate, the mutation operation is performed. The rule of variation about the number of hidden-layer neurons and learning rate is shown in Equation (10):

$$
\begin{aligned}
c.Ln &= \text{abs}(c.Ln + \text{random.randint}(-3, 3)) \\
c.lr &= \text{abs}(c.lr + \text{random.uniform}(-0.001, 0.001))
\end{aligned}
\tag{10}
$$

where $c.Ln$ is the number of hidden-layer neurons in a population and $c.lr$ is the learning rate.

*3.3. GA-LSTM Model*

In this paper, the GA and LSTM neural network models are combined to construct a ship trajectory prediction model based on GA-LSTM. Firstly, the GA is adopted to optimize the hyperparameters of the LSTM network, and then the best combination of learning rate and the number of hidden-layer neurons is obtained to further improve the nonlinear mapping ability of the model; in addition, the GA-LSTM model constructed by the optimal parameter combination is used as the nonlinear transformation function $f(\cdot)$ between the input sample and output sample; on this basis, the nonlinear transformation function $f(\cdot)$ is applied to obtain the position information of the ship at the next moment. The specific operation process of the model is as follows:

(1)    Selecting training data set.

In order to obtain the best-fitting effect of the function $f(\cdot)$, the input data of the GA-LSTM model are composed of the navigation information of the ship at the past three moments, which are represented by the vector $\boldsymbol{u}$, and the target output is represented by the vector $\boldsymbol{M}$. As shown in Formula (11):

$$
\begin{aligned}
\boldsymbol{u}_i &= [lat_{i-2}, lat_{i-1}, lat_i, lon_{i-2}, lon_{i-1}, lon_i, sog_{i-2}, sog_{i-1}, sog_i, cog_{i-2}, cog_{i-1}, cog_i]^T \\
\boldsymbol{M}_i &= \begin{bmatrix} lat_{i+1} \\ lon_{i+1} \end{bmatrix}
\end{aligned}
\tag{11}
$$

(2)    Optimizing LSTM network parameters with the GA.

a. Taking the learning rate and the number of hidden-layer units of LSTM model as the optimization objects, and then performing the initialization of the population and the chromosome encoding and decoding operations.

b. Calculating the fitness value of each individual in the initial population.

c. Performing selection, crossover, and mutation operations on chromosomes.

d. Decoding chromosomes and calculating the fitness of individuals in the population. The smaller the fitness value in this algorithm, the more the individual should be retained; otherwise, the individual should be eliminated.

e. If the genetic termination conditions are not met, it will return to Step c. If the genetic termination conditions are met, the optimal parameters calculated by the GA are taken as the final parameters of the LSTM network model.

(3)    Training the GA-LSTM model.

Inputting $\boldsymbol{u}$ and $\boldsymbol{M}$ into the GA-LSTM network of the optimal parameter combination, the output of the GA-LSTM network model is the position of the ship at the next moment. The difference between the target output $\boldsymbol{M}$ and the predicted output $f(\boldsymbol{u})$ of the model is represented by the error $\boldsymbol{e}$, which is $\boldsymbol{e} = \boldsymbol{M} - f(\boldsymbol{u})$. The GA-LSTM model minimizes the fitting error $E(\boldsymbol{e}^T\boldsymbol{e})$ according to the mean square error and finally obtains the best-fitting function $\widehat{f}(\cdot)$ between the input samples and the output samples.

(4)    Predicting ship trajectory.

The experimental data are sent to the GA-LSTM model, and then the ship navigation position at the next moment is calculated by using the best-fitting function $\widehat{f}(\cdot)$.

The ship trajectory prediction model framework is shown in Figure 2. The framework is mainly composed of three parts: data preprocessing, model analysis, and error analysis. Data preprocessing is an essential part of GA-LSTM model, and the data after preprocessing can improve the overall performance of the model to a certain extent. For model analysis, the GA is introduced into the selection of network hyperparameters based on the LSTM network model, which reduces the influence of artificial determination to some extent. For error analysis, visualization and index evaluation are used to further verify the feasibility and performance of the proposed method.

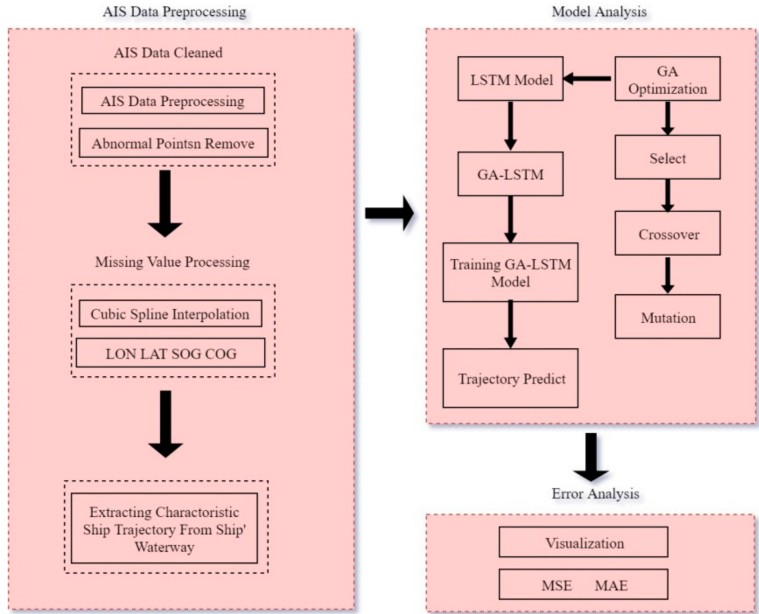

**Figure 2.** Flowchart of the vessel trajectory prediction framework.

## 4. Experiments and Analysis

### 4.1. Model Evaluation Index

In this paper, the overall performance of the model is evaluated through mean square error (MSE) [13] and mean absolute error (MAE) [12]. The smaller the value of MSE and MAE are, the higher the prediction accuracy is. The calculation formulas of MSE and MAE as follows:

$$MSE = \frac{1}{P}\sum_{i=1}^{p}(Y_i - y_i)^2 \tag{12}$$

$$MAE = \frac{1}{P}\sum_{i=1}^{p}|Y_i - y_i| \tag{13}$$

where $P$ is the total AIS data, $Y_i$ is the predicted value of the network model, and $y_i$ is the expected output value.

### 4.2. AIS Data Sources and Preprocessing

The AIS data were collected in November 2020, and the experimental area was from the Zhuankou waterway of the Wuhan section of the Yangtze River to the Baihushan crossing area. Because of the interrupted or missing data in the process of AIS signal sending, transmission, and reception and that the time series data with large deviations may appear in the AIS data, it is necessary to preprocess the data appropriately. The preprocessing process [33] of the collected AIS data in this paper is as follows:

Firstly, removing invalid data, mainly including:

(1)    MMSI is not a 9-bit data value.
(2)    AIS attribute information contains a large amount of data with null values.
(3)    In this paper, the LAT range of the track point is set to [110.00, 115.00], the LON range is set to [30.00, 32.00], the SOG range is set to [2.0–14.0], the SOG range is set to [0–360], and the distribution of research data after AIS data cleaning is shown in Figures 3 and 4.
(4)    Treatment of missing values.

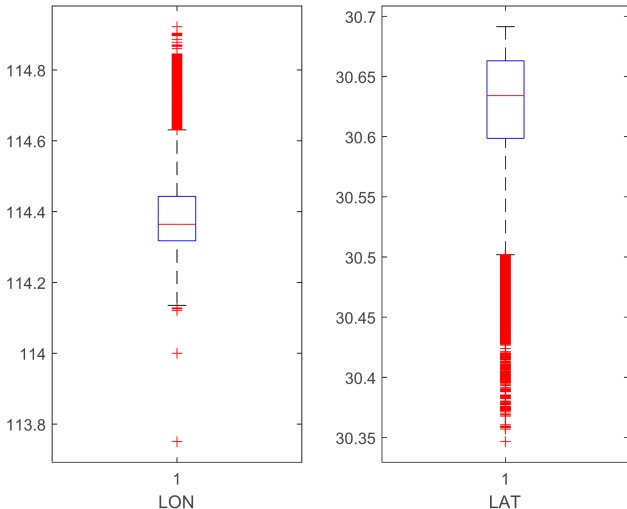

**Figure 3.** LON and LAT distributions.

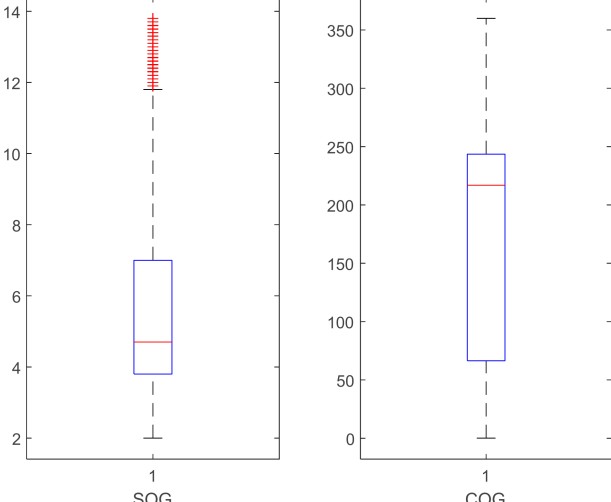

**Figure 4.** SOG and COG distributions.

In the experimental data in this paper, there are some missing AIS data for individual ships. As for the problem of missing values for ship LON and LAT, the method of cubic spline interpolation [34] is used to complete the model, as shown in Figure 5 for the interpolation results of completing missing values of LON and LAT data in two AIS data values. For the missing of SOG and COG during this period, considering the relatively constant characteristics of ship SOG and COG in a short period of time, the average value is used for interpolation.

(5) The attribute data contained in AIS information have different dimensions, so the trajectory data are normalized between 0 and 1. In this paper, the deviation method [3] is used for processing, and the normalization formula is shown in Equation (14):

$$X' = \frac{X - X_{\min}}{X_{\max} - X_{\min}} \tag{14}$$

where $X_{\min}$ is the minimum value in the experimental data, $X_{\max}$ is the maximum value, $X$ is the original data value, and $X'$ is the normalized data value.

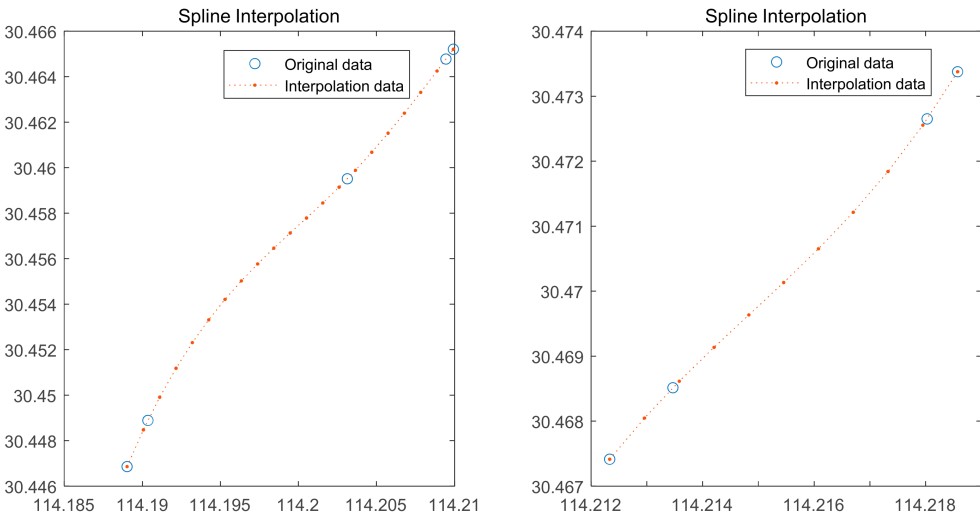

**Figure 5.** Cubic spline interpolation results.

### 4.3. Experimental Methods

After data preprocessing, a series of typical ship trajectories with large turning amplitudes is selected for experiments. In this paper, considering model calculation complexity, calculation time, and prediction performance for the SVM model, the radial basis function $K(x, y) = \exp(-||x - y||_2/\sigma^2)$ is selected as the kernel function, where the values of kernel function parameter $\sigma$ and penalty coefficient $C$ are $\sigma^2 = 3$ and $C = 50$, respectively. For the LSTM network model, 30 hidden-layer neurons are selected, the number of iterations is 1000, and the learning rate is 0.01. The parameter values for the GA [35,36] are shown in Table 1, where MaxGenerations is the maximum number of iterations before the GA stops optimization and PopulationSize is the size of the initial population. The smaller its value is, the more prone sick populations are to appear. The larger the population size is, the more difficult the algorithm is to converge and the lower robustness it has. CrossoverPop is the crossover probability; if the value is too large, it is easy to miss the optimal individual and the randomness is large, while the crossover probability is too small to effectively update the population. MutationPop is the mutation probability; if its value is too small, the diversity of population decreases too quickly, which easily leads to the loss of effective solutions and is hard to repair. If its value is too large, the probability of the optimal individual being destroyed also increases, which is not conducive to finding the optimal solution.

**Table 1.** Parameter values for GA.

| MaxGeneration | PopulationSize | CrossoverPop | MutationPop |
|---|---|---|---|
| 100 | 40 | 0.8 | 0.2 |

### 4.4. Visualized Comparative Analysis of Experimental Results

After data preprocessing, two typical trajectories are selected for experiments. This paper selects AIS data with MMSI values of 413826669 (ship-1) and 413997528 (ship-2) and compares the experiments with the LSTM model and the SVM model. The original trajectories of the two ships are shown in Figure 6.

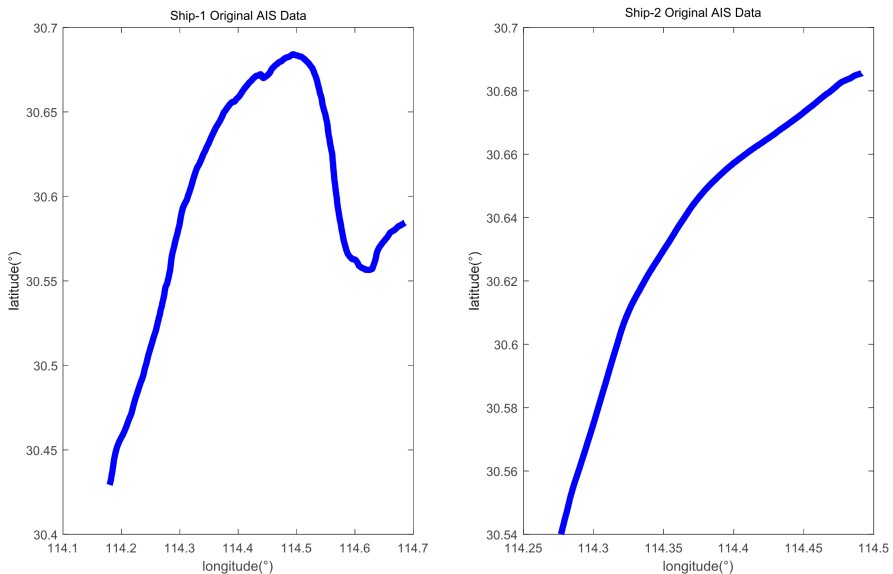

**Figure 6.** The original trajectories of experimental ships.

### 4.4.1. Visual Analysis of Ship-1 Trajectory Prediction

It can be seen from Figure 6 that ship-1 has a tendency of greater steering amplitude and continuous maneuvering. Firstly, the AIS data of ship-1, LOG, LAT, SOG and COG, are taken as the input samples of the GA-LSTM network model. The position of ship-1, LON and LAT, are taken as the output samples of the model, and the experimental results are shown in Figure 7.

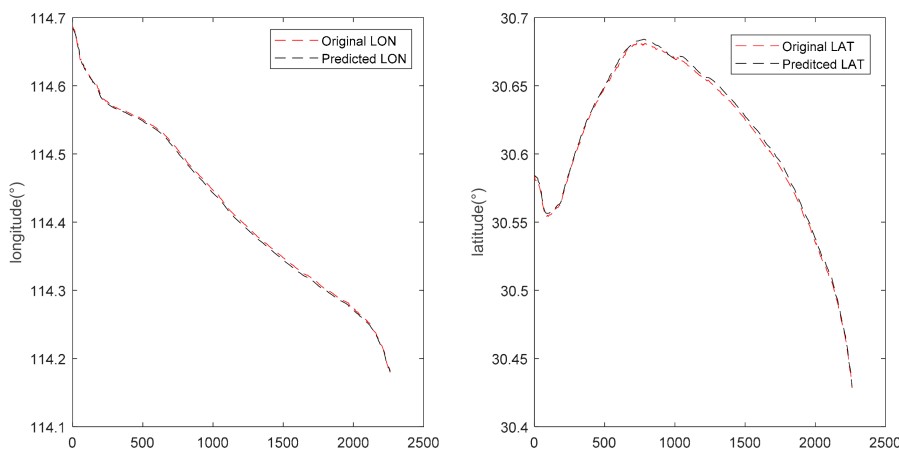

**Figure 7.** Prediction results of LON and LAT of ship-1.

The collected AIS data of ships are divided into training and test sets according to the above method. Figure 7 shows the LON and LAT of the ship predicted by the GA-LSTM model proposed in this paper. It can be seen that the LON and LAT predicted by this model are basically consistent with the actual LON and LAT of ship-1, which can accurately predict the position of ships in inland rivers.

In order to further prove the feasibility and effectiveness of the proposed model, this paper conducts a comparative analysis of the LSTM model and the SVM model. The experimental results are shown in Figure 8. It can be seen that: (1) For the location prediction of ship-1, the SVM model performs the worst; even if the predicted ship trajectory has the same general trend, the position information has a larger deviation. This is due to the weak generalization ability of the SVM model and its ease of falling into local extreme values. (2) The prediction effect of the LSTM method is slightly better than that of the SVM method; however, the two methods are not as good as the method proposed in this paper. This is because the network hyperparameters of the LSTM method have difficulty manually obtaining the optimal solution, which makes the model prediction performance lower than that of GA-LSTM method. (3) In order to better compare the pros and cons of the three methods, zoom in and analyze the position of ship-1′s navigation status: the LSTM and GA-LSTM methods have comparable predictive performance in ship-1′s direct navigation state (see ① in Figure 8). The GA-LSTM model shows better performance when performing large-scale steering or continuous steering (Figure 8 at ②). In general, the GA-LSTM model performs best and can predict ship-1′s sailing position more accurately. The reason is that the GA obtains a better combination of LSTM network parameters, which makes the method proposed in this paper better than the ship trajectory predicted based on the SVM and LSTM models.

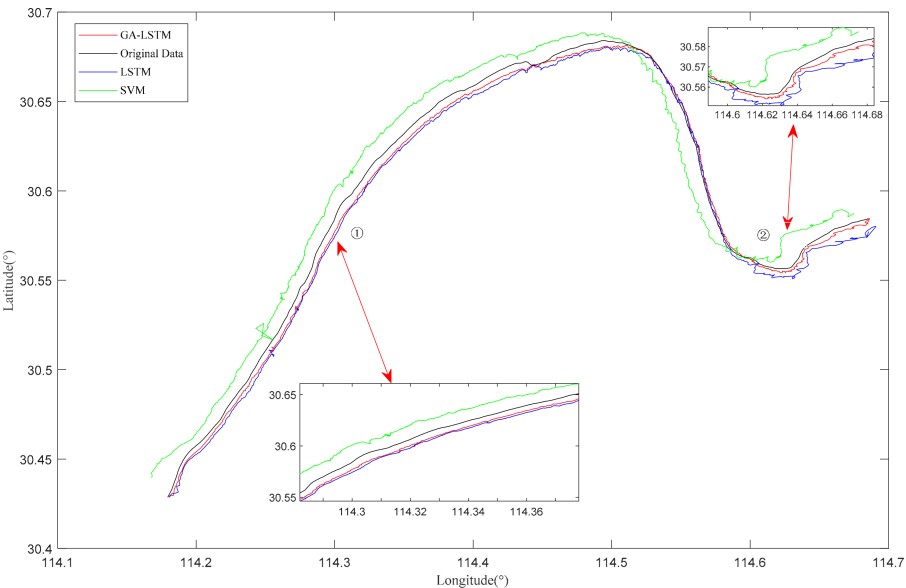

**Figure 8.** Comparison of experimental results of ship-1 trajectory prediction.

### 4.4.2. Visual Analysis of Ship-2 Trajectory Prediction

The AIS data of ship-2 are also used to predict the LON and LAT of the trajectory by the GA-LSTM model. The experimental results are shown in Figure 9. It can be seen from Figure 9 that the LON and LAT of ship-2 predicted by GA-LSTM are basically the same as the LON and LAT of the actual position in the position prediction of ship-2. Similarly, the trajectory was predicted by the SVM and LSTM models, and the experimental results are shown in Figure 10. It can be seen that the method proposed in this paper has higher prediction performance and better effect, and it can effectively predict the navigation position of ship-2, which further proves the effectiveness and feasibility of the method proposed in this paper.

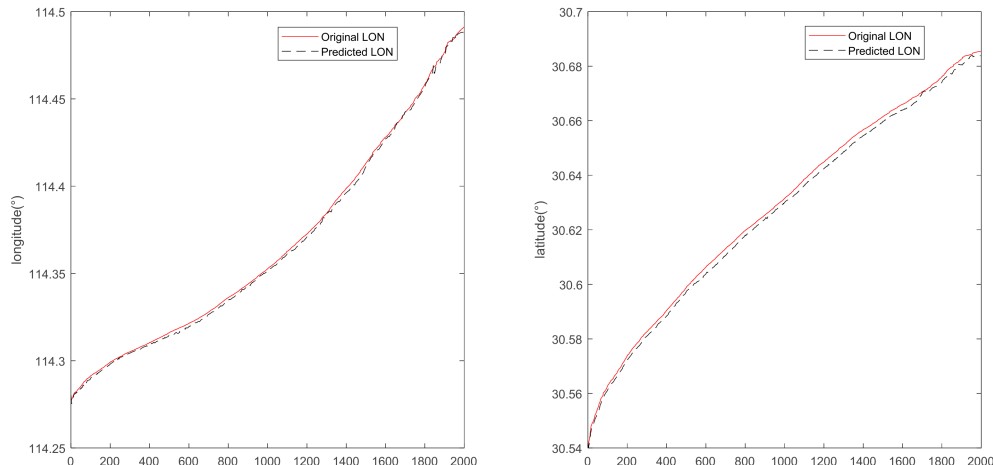

**Figure 9.** Prediction results of LON and LAT of ship-2.

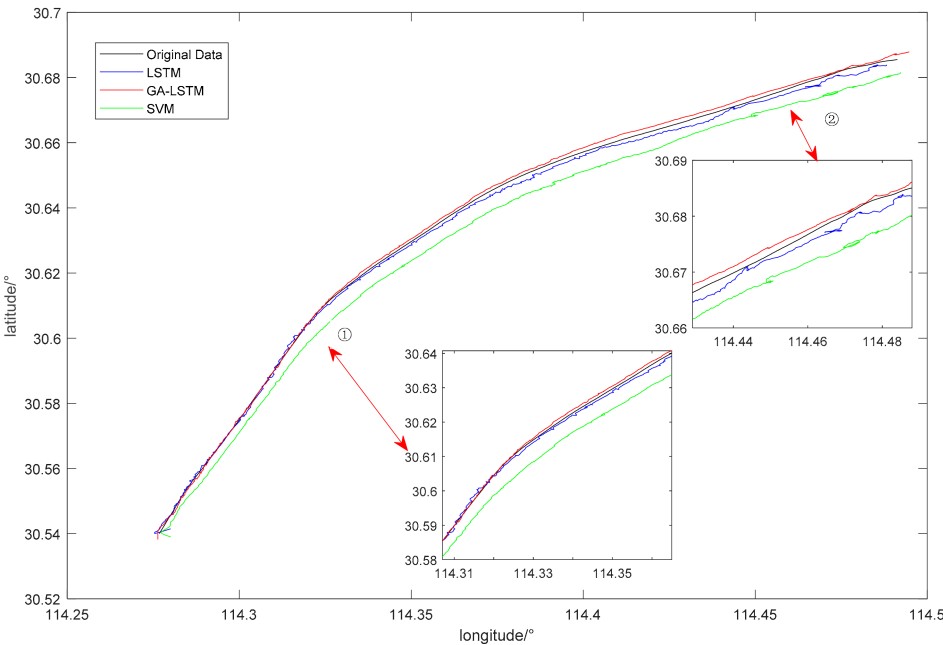

**Figure 10.** Comparison of experimental results of ship-2 trajectory prediction.

### 4.5. Model Performance Index Analysis

In order to further analyze the prediction effect of the GA-LSTM model on the two typical trajectories, this paper adopts MSE and MAE to evaluate the performance of the models. The index analysis results of the three methods are shown in Table 2.

It can be seen from Table 2 that the GA-LSTM model is the lowest in both evaluation indicators, its accuracy is relatively the highest, and its model prediction performance is better. For ship-1, when the optimal parameter combination is (11, 0.0165), the LAT MSE and MAE predicted by this method are $1.6393 \times 10^{-6}$ and 0.0014, respectively; meanwhile, when the optimal parameter combination for LON prediction is (7, 0.023), the MSE and MAE are $4.3188 \times 10^{-6}$ and 0.0024, respectively. For ship-2, the LAT MSE and MAE predicted with the optimal parameter combination (13, 0.0163) are $3.0375 \times 10^{-6}$ and 0.0017, respectively; the LON MSE and MAE predicted by the optimal parameter combination (14, 0.0105) are $1.8304 \times 10^{-6}$ and 0.0012, respectively. Both indicators are the lowest. It can be seen that firstly adopting the GA to optimize the key hyperparameters of the LSTM network model and then using the optimal parameter combination to construct

the GA-LSTM trajectory prediction model can effectively improve the performance and accuracy of prediction.

**Table 2.** Performance index analysis of models on 3 models.

| | Model | Position | MSE | MAE | Optimal Paramter Combination | |
|---|---|---|---|---|---|---|
| | | | | | Numb of Neuron | Learning Rate |
| ship-1 | SVM | LAT | $9.979 \times 10^{-6}$ | 0.002 | | |
| | | LON | $1.4957 \times 10^{-5}$ | 0.0034 | | |
| | LSTM | LAT | $2.6257 \times 10^{-6}$ | 0.0015 | | |
| | | LON | $7.8145 \times 10^{-6}$ | 0.0026 | | |
| | GA-LSTM | LAT | $1.6393 \times 10^{-6}$ | 0.0014 | 11 | 0.0165 |
| | | LON | $4.3188 \times 10^{-6}$ | 0.0024 | 7 | 0.0230 |
| ship-2 | SVM | LAT | $1.3404 \times 10^{-5}$ | 0.0035 | | |
| | | LON | $1.0037 \times 10^{-5}$ | 0.0029 | | |
| | LSTM | LAT | $5.4005 \times 10^{-6}$ | 0.0023 | | |
| | | LON | $8.742 \times 10^{-6}$ | 0.0027 | | |
| | GA-LSTM | LAT | $3.0375 \times 10^{-6}$ | 0.0017 | 13 | 0.0163 |
| | | LON | $1.8304 \times 10^{-6}$ | 0.0012 | 14 | 0.0105 |

### 4.6. Real-Time and Popularization Analysis of Model

The experiment was carried out using the Windows 10 system, the central processing unit was a 2.90 GHz i5 processor with a memory of 32.0 GB, and the experiment was carried out with MATLAB2021b. In the comparison experiment of this paper, the initial population of the GA algorithm was set to 40, and the maximum number of iterations was 100. The training time of the network and the execution time after training were used to compare and discuss the real-time analysis and generalization of the network. The experimental results are shown in Table 3.

**Table 3.** Comparison of real-time analysis of 3 models.

| | Model | Training Time T/S | | Execution Time T/S | |
|---|---|---|---|---|---|
| | | LON | LAT | LON | LAT |
| ship-1 | SVM | 55.915569 | 56.1377 | 0.426141 | 0.424251 |
| | LSTM | 53.9218 | 53.766166 | 0.403474 | 0.404457 |
| | GA-LSTM | 154.213806 | 166.647849 | 0.129239 | 0.129546 |
| ship-2 | SVM | 56.137049 | 56.586795 | 0.427225 | 0.433015 |
| | LSTM | 53.671202 | 53.841451 | 0.400593 | 0.401865 |
| | GA-LSTM | 145.977239 | 142.860358 | 0.117013 | 0.123151 |

From the comparative analysis in Table 3, we can see that the methods adopted in the experiment require a certain amount of time to construct the network, and the difference in network structure makes the time used for network training different. It can be seen from Table 3 that: ① In the network training stage, the training times of the LSTM method and the SVM method are equivalent; after adding GA, the time consumption of the network training stage is increased by about two times; ② In the network execution stage, the time consumptions of the LSTM and SVM methods are basically the same. Compared with the other two algorithm models, the time consumption of the GA-LSTM method is not significantly increased but decreases slightly. In the MATLAB2021b environment, the execution time of GA-LSTM is about 0.2 s.

It can be seen that in practical applications, except for about 3 min in the program start-up training phase, the ship trajectory prediction results can be obtained quickly in other time periods. If another compiled language such as C writes the algorithm into the hardware to

run, the running speed will be further improved. Therefore, the method proposed in this paper can meet the needs of certain scenarios in terms of real-time performance and has a good generalization.

## 5. Conclusions

In the background of world economic globalization, shipping has become one of the most important modes of transportation in international trade. The number, types, and new routes of ships continue to increase. Although the shipping trade shows a thriving atmosphere, it also makes the channel congested and the load increases, which affects the safety of ship navigation and seriously threatens the life and property safety of ship personnel. From the analysis of the ship accident investigation organization, it can be seen that human error is the main cause of marine and inland river accidents. The key to the safe navigation of ships lies in the perception of the surrounding navigation environment during the navigation process and the effective use of varied information for correct analysis and decision making. As a common navigation environment perception means, AIS has some deficiencies in the process of receiving and sending ship information, which restricts the maneuvering behavior of the ship. Knowing how to use the AIS information to accurately predict the trajectories of their own ship and the target ship in a specific time is vital for a ship driver to make a correct evaluation and decision.

In order to improve the prediction accuracy and calculation efficiency when predicting the future position of the ship in a specific period, this paper introduces the LSTM network and the GA optimization algorithm to the future position prediction of the ship and proposes a new method based on GA-LSTM to predict the course and position of inland ships. Considering the disadvantages of AIS equipment in the process of receiving and sending ship information, firstly, the collected real-time AIS data are preprocessed and the cubic spline interpolation method is adopted to interpolate data for the data loss of individual ships. Then, RNNs have powerful time-series processing capabilities, and are able to use historical information to accurately predict future state, building the ship trajectory prediction model based on the LSTM network model, and utilizing the GA optimizes the hyperparameters of the LSTM network. The GA-LSTM prediction model can minimize the impact of hyperparameter factors on the accuracy of ship trajectory prediction. In this paper, two typical ship trajectories in the Wuhan section of the Yangtze River are selected for prediction experiments and compared with the classic SVM method and the LSTM method. The experimental results show that the GA-LSTM model proposed in this paper has higher prediction accuracy and prediction speed; this model not only performs well in predicting the trajectory of the ship sailing in a straight line but also has a strong advantage in trajectory prediction when the ship starts to maneuver or maneuvers at a large angle. On the basis of high computing efficiency, it can predict ship trajectory in real time and accurately provide effective guarantee measures for the safe navigation of ships, and it has better generalization ability.

This model solves the problems of low prediction accuracy and complex calculation of ship trajectory prediction to a certain extent and has good practical application value in the intelligent navigation of inland river ships. The GA-LSTM model is similar to other time-series models, and the effect of ship trajectory prediction with long-term information is not ideal. On the basis of ensuring the prediction accuracy and speed, if the accuracy of long-distance ship position prediction can be improved, it will provide great help for ship collision avoidance and other maneuvering processes. Even though the model in this paper is based on a recursive network model, the calculation cost is relatively high in long-distance ship position prediction, and it is not practical in the collision avoidance maneuvering of unmanned ships. How to predict the ship's position without losing prediction accuracy and efficiency and how to decrease the computational cost as well will be carried out in further work.

**Author Contributions:** Conceptualization, Y.Z.; Data curation, L.L. and D.Z.; Formal analysis, L.Q.; Funding acquisition, Y.Z. and C.Z.; Investigation, Y.Z.; Methodology, L.Q.; Resources, Y.M. All authors have read and agreed to the published version of the manuscript.

**Funding:** The research is financially supported by National Nature Science Foundation of China (51979215; 52171349; 52171350).

**Institutional Review Board Statement:** Not applicable.

**Informed Consent Statement:** Not applicable.

**Data Availability Statement:** Not applicable.

**Conflicts of Interest:** The authors declare that they have no known competing financial interests or personal relationships that could have appeared to influence the work reported in this paper.

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
