# Peer review of "A New Method of Inland Water Ship Trajectory Prediction Based on Long Short-Term Memory Network Optimized by Genetic Algorithm"

_applsci, doi:10.3390/app12084073_

Round 1
Reviewer 1 Report
Authors have proposed in this article "A new method of inland water ship trajectory prediction based on long short term memory network optimized by genetic algorithm"
1.The authors should clearly state this article contribution to be to the body of knowledge
2. The literature study must be improve by adding more literature of the ship navigation system.
3. There are few grammatical errors in abstract and body which must be corrected. and functions definition in equation 2 must be defined:
f (. )
Author Response
Response to Reviewer 1 Comments
Point 1: The authors should clearly state this article contribution to be to the body of knowledge.
Response 1: Thanks to the reviewer for their comments, the revision of point 1 is as follows:
The contribution of this paper to the body of knowledge is added in the conclusion section of this paper. It mainly solves the problems of low accuracy and real-time performance of ship trajectory prediction, provides more effective guarantee measure for the safe navigation of ships, and has a good application prospect in the research of ship trajectory prediction. It also provides a new method for the intelligentization of inland ships in the future.
Point 2: The literature study must be improved by adding more literature of the ship navigation system
Response 2: Thanks to the reviewer for their comments, the revision of point 2 is as follows:
There are few references cited in this paper, and the literature related to ship navigation system are added in the revised manuscript, with a total of 36 references. The specific reference information is shown in the revised manuscript.
Point 3: There are few grammatical errors in abstract and body which must be corrected. and functions definition in equation 2 must be defined: f(·).
Response 3: Thanks to the reviewer for their comments, the revision of point 3 is as follows:
The grammatical errors in this paper have been corrected comprehensively in the revised manuscript, and language polish has been made. The formulas in the revised manuscript have been re-edited.

Reviewer 2 Report
In this paper, long short term memory network (LSTM) is used to establish an inland river ship trajectory prediction model. The genetic algorithm (GA) is proposed to optimize the key hyperparameters of LSTM. The paper is readable and easy to understand. The whole paper becomes well organized. However, the review has three minor concerns.
- In general, the edited equations should be Improved on its presentation.
- There are many parameters for the Genetic Algorithm (GA), but the impact of different parameter settings on algorithm performance is missed. One paragraph or one table should be added with more information about GA parameters. What is the crossover probability? What is the mutation rate value, the population size....The choice of parameters values is not justified. Is it based on other experiments presented in the literature? Which ones? Furthermore, the parameters tunning can provides better results.
- Some recent studies using neural networks reported satisfactory results when the vessel moves along a straight trajectory. However, the main problems appear when the ship starts to maneuver, where has been detected a large discrepancy between the results of their simulated and real data. This issue does not occur here. Is this fact related to the introduction of optimized hyperparameters obtained from GA?
Please spell check to eliminate some grammatical errors. Some minor comments are listed below.
- Page 1 line 1, In the abstract:
“
“Ship position prediction plays the key role to early warning and safety.” could be changed to “Ship position prediction plays a key role in the early warning and safety.”.
- Page 1, line 3, in the abstract: “…own ship and target ship in a specific period of time when …” could be changed “…
their ship and target ship in a specific time period when …”
- Page1, line 3, in the introduction “
volume, the activity of waterway transportation…” could be changed “
volume, the activity of the waterway transportation...”
- Page2, line 6: “Jiang [6] proposed to use the polynomial Kalman filter method to fit …” could be changed to “Jiang [6] proposed the polynomial Kalman filter method to fit …”
- Page 2, In the introduction, line 57: “In order to predict the ship trajectory quickly and accurately, this paper adopts LSTM network model as the technical basic to establish …” could be changed to “In order to predict the ship trajectory quickly and accurately, this paper adopts LSTM network model as the technical basis to establish …”
Author Response
Response to Reviewer 2 Comments
Point 1: In general, the edited equations should be Improved on its presentation.
Response 1: Thanks to the reviewer for their comments, the revision of point 1 is as follows:
The formula may be irregular due to different version of Office. The formula in the revised manuscript has been re-edited.
Point 2: There are many parameters for the Genetic Algorithm (GA), but the impact of different parameter settings on algorithm performance is missed. One paragraph or one table should be added with more information about GA parameters. What is the crossover probability? What is the mutation rate value, the population size....The choice of parameters values is not justified. Is it based on other experiments presented in the literature? Which ones? Furthermore, the parameters tunning can provides better results
Response 2: Thanks to the reviewer for their comments, the revision of point 2 is as follows:
There are many parameters for GA. A table is added in the revised manuscript to show the parameter values for GA. The setting rules are based on a lot of experiments and experience, and refer to the parameter settings in the literature, as shown in Table 1. Where, MaxGenerations is the maximum number of iterations before GA stops optimizating; PopulationSize is the size of the initial population. The smaller its value is, the more prone to appear sick populations. The larger the population size is, the algorithm is difficult to converge and has low robustness; CrossoverPop is the crossover probability, if the value is too large, it is easy to miss the optimal individual and the randomness is large, while the crossover probability is too small to effectively update the population; MutationPop is the mutation probability, If its value is too small, the diversity of the population will decrease too quickly, which will easily lead to the loss of effective solutions and it is not easy to repair. If its value is too large, the probability of the optimal individual being destroyed will also increase, which is not conducive to finding the optimal solution.
Table 1. Parameters value of GA
|
MaxGenerations |
PopulationSize |
CrossoverPop |
MutationPop |
|
100 |
40 |
0.8 |
0.2 |
Point 3: Some recent studies using neural networks reported satisfactory results when the vessel moves along a straight trajectory. However, the main problems appear when the ship starts to maneuver, where has been detected a large discrepancy between the results of their simulated and real data. This issue does not occur here. Is this fact related to the introduction of optimized hyperparameters obtained from GA?
Response 3: Thanks to the reviewer for their comments, the revision of point 3 is as follows:
Ship trajectory prediction is a relatively classic problem, and the current popular prediction models are generally constructed based on ships sailing in a straight line. However, for the prediction of the initial sailing state of the ship or the large-scale maneuvering, the results are not satisfactory, mainly because the ship is greatly affected by hydrology, meteorology and other environment when it starts sailing. The nonlinear mapping ability of neural network can learn deep-level features of samples based on a large amount of sample data. The purpose of learning is to find the optimal hyperparameters of the network. Therefore, the network structure is greatly affected by the hyperparameters of the model. Different hyperparameters make the prediction performance of the network have obvious differences. In the experiment of this paper, two typical ship trajectories in the research waters are selected for the experiment. The results show that the GA-LSTM model not only performs well for the trajectory prediction of the ship sailing in a straight line, but also for the trajectory prediction when the ship starts to maneuver or maneuvers at a large angle. It also shows strong advantages. Therefore, it can be seen that the GA-LSTM model established by GA optimizing the hyperparameters of LSTM has better trajectory prediction performance, and can predict the ship's trajectory in real time under different working conditions, which further guarantees the safe of ship navigation.
Meanwhile, the grammatical errors in this paper have been corrected comprehensively in the revised manuscript, and language polish has been made.
